# WHEN UNCERTAINTY, COVERAGE, AND REPRESENTATION MATTER IN ACTIVE LEARNING FRAMEWORKS

## ABSTRACT

Active learning (AL) aims to reduce annotation costs by querying informative and representative samples for labeling. Despite significant progress, many AL methods remain heuristic and lack a unified theoretical foundation. We present the first systematic and theory-guided framework that connects the core principles of AL (uncertainty, representation, and coverage) to a decomposition of generalization error into empirical risk, distributional discrepancy, model complexity, and confidence. This mapping not only explains why different AL strategies excel under varying annotation budgets but also provides a blueprint for designing future methods, positioning our work as a foundation for principled AL development. Our analysis unifies prior empirical observations into a generalization-theoretic foundation, complemented by extensive experiments on CIFAR and ImageNet subsets with self-supervised embeddings and pretrained encoders. Results show that representation is critical in early rounds to address cold-start issues; coverage promotes diversity in mid-budget regimes; and uncertainty becomes most effective once decision boundaries are partially learned. We also observe that while per-sample reductions in model complexity are modest, their cumulative effect across acquisition rounds is substantial. We further assess runtime behavior, highlighting trade-offs between theoretical alignment and scalability. Rather than proposing a new method, our contribution is a unifying and generalizable framework that explains strategies and guides principled AL design, bridging theory and practice.

## 1 INTRODUCTION

Active learning (AL) aims to select the most informative and representative samples from a large pool of unlabeled data to achieve high model performance with minimal annotation effort Settles (2009). This is especially important in domains like medical and satellite imaging, autonomous driving, and anomaly detection in IoT, where labeling is costly, time-consuming, and often requires expert input. In such settings, AL helps reduce annotation costs by prioritizing valuable samples. Timely selection of a well-curated subset can also accelerate model improvement and enable faster intervention Hacohen et al. (2022).

Numerous AL algorithms have been proposed to reduce annotation costs while maintaining high predictive performance Gal et al. (2017); Kirsch et al. (2019); Ash et al. (2021); Hacohen et al. (2022); Yehuda et al. (2022); Bae et al. (2024). Although many perform well in specific regimes, their effectiveness is typically supported only by empirical evidence without systematic connection to generalization theory. A deeper theoretical understanding of why certain approaches are effective under particular conditions and how they relate to fundamental learning principles remains limited.

This work addresses these gaps by revisiting sample selection through the lens of empirical risk minimization (ERM) Vapnik et al. (1974) and generalization bounds. We decompose the error into four components: **empirical risk**, **model complexity** via Rademacher complexity, **distributional discrepancy** between labeled and unlabeled data, and a **confidence** term. We map core AL principles of **uncertainty**, **coverage**, and **representation** to these components, showing how each strategy implicitly targets a distinct term. This explains the success of AL methods, even when these principles are not explicitly encoded. Notably, we show that although per-sample reductions in Rademacher complexity are modest, their cumulative effect across acquisition rounds is substantial, providing

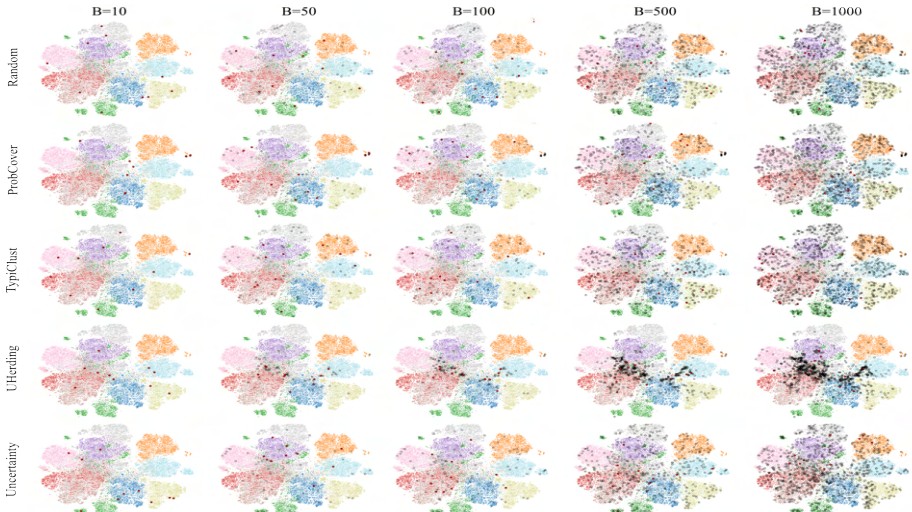

Figure 1: t-SNE visualization of sample selection behavior across different AL strategies on CIFAR-10 at growing annotation budgets (B). Points are projected into 2D using self-supervised embeddings and colored by ground-truth class. Black points indicate previously labeled samples; Reds denote newly selected points in the current round. Representation methods (e.g., TypiClust) emphasize broad class diversity early on, while uncertainty methods focus on ambiguous regions. As the budget increases, strategies diverge in prioritizing between uncertainty, representation, and coverage.

the first systematic mapping from generalization error decomposition to AL principles, and offering a foundation for principled method selection and design.

We propose a theory-inspired framework for understanding and predicting the behavior of diverse AL strategies across varying budget regimes. Beyond theoretical analysis, we examine how methodological complexity and computational scalability impact performance on datasets of differing complexity, highlighting practical trade-offs. Extensive experiments across datasets, budgets, and selection strategies, including uncertainty-, coverage-, and representation-based methods, validate our insights. We also address real-world concerns such as efficiency and scalability beyond standard benchmarks. Rather than introducing another AL algorithm, we aim to offer a **theory-driven framework** that guides when, why, and how different AL methods are likely to succeed or fail, supporting more informed method selection, evaluation, and design.

To illustrate the practical motivations underlying our framework, Figure 1 shows t-SNE visualizations of sample selection patterns for various AL methods at increasing annotation budgets on CIFAR-10. In the low-budget regime, representation-based methods like TypiClust Hacohen et al. (2022) promote broad class diversity by selecting one representative per cluster, while uncertainty-based methods such as UHerding Bae et al. (2024) and Uncertainty sampling Lewis (1995) target ambiguous regions, often neglecting some classes. As the budget grows, TypiClust continues sampling from dense areas but underexplores uncertain regions, whereas uncertainty-based methods broaden their coverage. ProbCover Yehuda et al. (2022) balances class coverage and local density but can overemphasize dense clusters, while random sampling yields uniform coverage throughout. These patterns show how different strategies implicitly prioritize uncertainty, representation, or coverage, and how the relevance of each principle shifts with budget. Such trade-offs highlight the need for theoretical frameworks to explain AL behavior across regimes.

## 2 RELATED WORK

**Uncertainty** methods are among the earliest and most studied AL strategies. Techniques like margin sampling Scheffer et al. (2001), entropy-based acquisition Settles (2009), and general uncertainty sampling Lewis (1995) prioritize low-confidence samples, assuming they are most informative for refining decision boundaries. Recent methods like BALD Kirsch et al. (2019) use Bayesian model

disagreement, while BAIT Ash et al. (2021) minimizes expected Bayes risk. These approaches are particularly effective in high-budget regimes, where model uncertainty estimates are more reliable.

**Representation**-based methods capture the data distribution by selecting diverse, representative samples. Classical approaches include $k$-means clustering Xu et al. (2003), medoids Aghaee et al. (2016), and median-based sampling Voevodski et al. (2012). Recent methods like TypiClust Hacohen et al. (2022) combine clustering with typicality to identify central points. These strategies are most effective in low-budget regimes, where uncertainty estimates are unreliable.

**Coverage**-based methods promote broad exploration of the input space using geometric or probabilistic criteria. Core-set selection Sener & Savarese (2017) minimizes the maximum distance between unlabeled points and their nearest labeled neighbors, while ProbCover Yehuda et al. (2022) extends this with probabilistic coverage. These methods perform well in low- to mid-budget regimes by bridging the benefits of spatial diversity and coverage.

**Hybrid** approaches combine multiple selection criteria to improve robustness. For instance, DBAL Gal et al. (2017) employs Bayesian dropout for uncertainty estimation while encouraging diversity. UHerding Bae et al. (2024) merges uncertainty with spatial coverage. Methods like semantic core-set selection Maalouf et al. (2022); Liang et al. (2024) and BADGE Ash et al. (2019) blend uncertainty and representation cues, often excelling in mid- to high-budget regimes.

Recent work has also attempted to evaluate AL methods from a generalization-oriented perspective. PALM Machnio et al. (2025) introduces a performance analysis framework to track AL dynamics as the labeled pool grows, using four empirical criteria: achievable accuracy, coverage efficiency, early-stage performance, and scalability. While such frameworks provide valuable empirical diagnostics, our work complements them by offering a theoretical perspective grounded in generalization error decomposition, helping to explain why different AL strategies succeed under varying conditions.

## 3 METHODS

We analyze AL through the lens of statistical learning theory, grounded in ERM and bounded by the Probably Approximately Correct (PAC) framework Valiant (1984). Our analysis identifies three core principles underlying effective AL strategies: uncertainty, representation, and coverage. Each principle primarily influences a different component of the generalization error and contributes variably depending on the annotation budget and dataset complexity. This perspective enables a principled understanding of the strengths and limitations of diverse AL methods.

### 3.1 PROBABLY APPROXIMATELY CORRECT LEARNING

PAC learning formalizes the notion of learnability, specifying whether a target concept can be learned from a finite number of samples with high confidence and accuracy, and quantifying the number of samples required. Let $\mathcal{X}$ denote the input space, $\mathcal{Y}$ the label space, and $\mathcal{C}$ a concept class of labeling functions $c : \mathcal{X} \to \mathcal{Y}$. In a realizable PAC setting, the learner observes $m$ i.i.d. samples $S = \{(x_i, y_i)\}_{i=1}^{m}$ where each $x_i$ is drawn from a distribution $\mathcal{D}$ over $\mathcal{X}$, and $y_i = c(x_i)$ is deterministically derived for some unknown target concept $c \in \mathcal{C}$. The learner then selects a hypothesis $h \in \mathcal{H}$, where $\mathcal{H}$ is the hypothesis class.

**Definition:** A class $\mathcal{H}$ is PAC-learnable if there exists an algorithm $\mathcal{A}$ such that, for all $\varepsilon, \delta \in (0, 1)$ and all target functions $c \in \mathcal{C}$, the output hypothesis $h = \mathcal{A}(S)$ satisfies

$$\mathbb{P}_{S \sim \mathcal{D}^m} [R(h) \leq \varepsilon] \geq 1 - \delta, \tag{1}$$

where the generalization error is defined as

$$R(h) = \mathbb{P}_{x \sim \mathcal{D}}[h(x) \neq c(x)]. \tag{2}$$

Hence, PAC learning is interpreted as probably (with probability at least $1 - \delta$) approximately correct (the hypothesis $h$ has error at most $\varepsilon$), aiming to characterize the sample complexity required to learn under uncertainty. In the agnostic PAC setting or with noisy stochastic labels, where $\mathcal{D}$ is defined over $\mathcal{X} \times \mathcal{Y}$ and loss is computed over $(x, y)$, the generalization error becomes

$$R(h) = \mathbb{P}_{(x,y) \sim \mathcal{D}}[h(x) \neq y]. \tag{3}$$

## 3.2 EMPIRICAL RISK MINIMIZATION

ERM is a learning principle or method to achieve PAC learning. It is a strategy for selecting a hypothesis from $\mathcal{H}$ by minimizing the loss over a finite set of training samples

$$\hat{R}_S(h) = \frac{1}{m} \sum_{i=1}^{m} \ell(h(x_i), y_i), \tag{4}$$

where $\ell$ is a loss function measuring the error between the true label $y_i$ and the prediction $\hat{y}_i = h(x_i)$. The ERM hypothesis is chosen as

$$h_S = \arg\min_{h \in \mathcal{H}} \hat{R}_S(h). \tag{5}$$

Since ERM does not guarantee generalization and may lead to overfitting, its effectiveness depends on the hypothesis class $\mathcal{H}$ having limited complexity.

## 3.3 GENERALIZATION BOUNDS

Generalization theory guarantees the effectiveness of ERM. It studies how well the empirical performance hypothesis on training data reflects true performance on unseen data. The general bound (uniform convergence) states that with high probability over the sample $S \sim \mathcal{D}^m$

$$\sup_{h \in \mathcal{H}} |R(h) - \hat{R}_S(h)| \leq \varepsilon(m, \mathcal{H}, \delta), \tag{6}$$

where $R(h) = \mathbb{E}_{(x,y) \sim \mathcal{D}}[\ell(h(x), y)]$ is the true risk and $\hat{R}_S(h)$ is the empirical risk. In the case of binary classification with a hypothesis class $\mathcal{H}$ of VC dimension $d$, the generalization error deviation bound for a fixed $m$ is

$$\mathbb{P}\left[\forall h \in \mathcal{H} : |R(h) - \hat{R}_S(h)| \leq \varepsilon(m, d, \delta)\right] \geq 1 - \delta, \tag{7}$$

where

$$\varepsilon(m, d, \delta) = \mathcal{O}\left(\sqrt{\frac{d + \log(1/\delta)}{m}}\right), \tag{8}$$

which indicates that ERM is effective when the hypothesis class has low complexity and sufficient data is available. A hypothesis with small empirical error is likely to have small true error if $\mathcal{H}$ is not too expressive.

To determine how many samples $m$ are sufficient to achieve a desired generalization guarantee with accuracy $\varepsilon$ and confidence $1 - \delta$, one can use probabilistic sample complexity bounds as

$$\mathbb{P}\left[\sup_{h \in \mathcal{H}} |R(h) - \hat{R}_S(h)| > \varepsilon\right] \leq \delta, \tag{9}$$

where

$$m = \mathcal{O}\left(\frac{d + \log(1/\delta)}{\varepsilon^2}\right). \tag{10}$$

## 3.4 ACTIVE LEARNING WITH PAC GUARANTEES

In AL, the learner is not limited to a fixed random sample but can actively select which examples to label. The goal is to achieve a desired generalization error with fewer labeled samples by querying the most informative instances from an oracle (e.g., a human annotator). The most common AL setting is pool-based, where the learner selects instances from a pool of unlabeled data, queries their labels, and updates its hypothesis incrementally. This adaptive, non-i.i.d., query-driven sampling process aims to achieve PAC-style generalization guarantees with reduced annotation effort.

Let $\Lambda(\varepsilon, \delta, \mathcal{H})$ denote the number of label queries (label complexity) required to learn a hypothesis $h$ such that $\mathbb{P}[R(h) \leq \varepsilon] \geq 1 - \delta$. AL is effective when $\Lambda$ is asymptotically smaller than the sample complexity $m$ of passive learning (Eq. 10). In noiseless settings, many AL strategies aim to reduce the version space–the set of hypotheses consistent with the labeled data–by querying points in regions of disagreement, where candidate hypotheses yield conflicting predictions. A disagreement coefficient Hanneke (2007) is defined to quantify the efficiency of this uncertainty reduction, enabling PAC bounds on expected label complexity.

## 3.5 GENERALIZATION ERROR FOR ACTIVE LEARNING

Let $Z = (X, Y)$ denote the joint random variables over input and label spaces in supervised learning, with $X : \Omega \to \mathcal{X} \subseteq \mathbb{R}^n$ and $Y : \Omega \to \mathcal{Y} \subseteq \mathbb{R}$. The joint distribution factorizes as $P_Z = P_X P_{Y|X}$, where $P_X$ is the marginal over inputs and $P_{Y|X}$ is the conditional label distribution. As minimizing the true risk is typically intractable, the learning objective is relaxed to empirical risk minimization over a finite sample, with constraints on model complexity. AL aims to learn a hypothesis $h \in \mathcal{H}$ that minimizes the true risk using as few labeled examples as possible. Unlike passive learning, where training data is drawn i.i.d. from $P_X P_{Y|X}$, AL selects data from an induced distribution $P_Q P_{Y|X}$, where $P_Q$ reflects the marginal over queried inputs. The goal of an optimal query strategy is to choose $P_Q$ such that the resulting training set achieves low generalization error.

The sampling shift introduces a distribution mismatch between $P_Q$ and the true data distribution $P_X$, analogous to the covariate shift in domain adaptation, where generalization bounds decompose the target risk into the source error, a divergence term between source and target distributions, and an irreducible joint error Ben-David et al. (2010). Similarly, AL can be analyzed within the ERM framework under sampling bias Menden et al. (2025), by deriving a PAC-style generalization bound for the AL risk that incorporates both distributional shift and model complexity as

$$\underbrace{R(h)}_{\text{True Risk}} \leq \underbrace{\hat{R}_S(h)}_{\text{Empirical Risk}} + \underbrace{d_{\mathcal{F}}(P_X, P_Q)}_{\text{Distribution Discrepancy}} + \underbrace{2\operatorname{Rad}(\ell \circ \mathcal{H} \circ S)}_{\text{Model Complexity}} + \underbrace{\alpha\sqrt{\frac{2\log(4/\delta)}{m}}}_{\text{Confidence Term}}, \qquad (11)$$

where $d_{\mathcal{F}}$ is an integral probability metric (IPM) over a class of measurable functions $\mathcal{F}$ constrained by bounded complexity Müller (1997), and $\alpha > 0$ is a constant tied to the Lipschitz continuity of the loss function. The term $\operatorname{Rad}(\cdot)$ denotes the Rademacher complexity of the loss function over the $m$ labeled samples $S$ for hypotheses $h \in \mathcal{H}$, and $\delta > 0$ is a confidence parameter. This bound holds with probability at least $1 - \delta$ over the random draw of $S$.

We analyze each generalization term in Eq. 11 below and show how they relate to the core AL selection criteria: uncertainty, representation, and coverage.

**Empirical Risk** The empirical risk captures the informativeness of labeled samples. While the true risk reflects the expected loss over the full data distribution, it is typically intractable due to the unknown nature of $P_Z$. Instead, learning algorithms minimize the empirical risk, which estimates the expected loss over a labeled dataset $S$ as defined in Eq. 4. In classification tasks with cross-entropy loss, the empirical risk takes the form

$$\hat{R}_S(h) = -\frac{1}{m}\sum_{i=1}^{m} y_i^T \log h(x_i), \qquad (12)$$

where $y_i$ is the one-hot encoded label and $h(x_i)$ is the predicted probability output by the model. Selecting informative samples that reduce empirical risk under this formulation encourages the model to refine its decision boundaries. This makes the empirical risk term central to uncertainty-based strategies in AL. In particular, querying samples with high loss, uncertainty, or low confidence is likely to reduce empirical risk upon retraining. Hence, minimizing this term aligns with prioritizing informative samples, those that the model is currently uncertain about or prone to misclassifying.

**Distribution Discrepancy** The distribution discrepancy term quantifies the representativeness and diversity of the queried samples. It measures how closely the empirical distribution $P_Q$, induced by the labeled set $S$, approximates the true data distribution $P_X$, and is given by

$$d_{\mathcal{F}}(P_X, P_Q) = \sup_{f \in \mathcal{F}} \left| \mathbb{E}_{x \sim P_X}[f(x)] - \mathbb{E}_{x \sim P_Q}[f(x)] \right|. \qquad (13)$$

A smaller discrepancy indicates that the labeled samples provide a faithful approximation of the input space, making this term central to representation-based AL strategies. Minimizing the discrepancy encourages the selection of diverse and representative samples that adequately span the data manifold. This becomes especially important in low-budget regimes, where uncertainty estimates are often unreliable due to limited labeled data. In such cases, AL algorithms should prioritize sampling from underrepresented or low-density regions of the input space to reduce discrepancy, thereby improving the representativeness of the labeled set.

**Model Complexity**    Model complexity is influenced by how effectively the labeled samples constrain the hypothesis class over the input space. Selecting diverse and non-redundant samples enhances coverage and exploration, thereby reducing the model's effective capacity. In contrast to empirical risk and distributional discrepancy, which are more directly impacted by the selected samples, the Rademacher complexity captures a subtler indirect effect: it quantifies the richness of the function class that remains consistent with the labeled data. Although not specific to individual samples, it is shaped by the diversity and spread of the labeled set. The effective capacity is often measured using the empirical Rademacher complexity Bartlett & Mendelson (2002) as

$$
\text{Rad}(\mathcal{K}) = \mathbb{E}_\sigma \left[ \sup_{k \in \mathcal{K}} \frac{1}{m} \sum_{i=1}^m \sigma_i \ell(h(x_i), y_i) \right], \tag{14}
$$

where $\mathcal{K} = \ell \circ \mathcal{H} \circ S$ denotes the composition of the loss function, hypothesis class, and labeled dataset, and $\sigma_i \in \{\pm 1\}$ are Rademacher variables. Lower Rademacher complexity implies better generalization, highlighting the importance of sample diversity in controlling model capacity.

Adding new labeled points from underexplored regions of the input space increases coverage and distributional spread, thereby reducing the supremum in Rademacher complexity. While the per-sample reduction is typically modest, its cumulative effect across acquisition rounds can be significant, leading to progressively tighter generalization bounds (as empirically shown in Sections *Budget Size* and *Dataset Complexity*). Thus, actively selecting samples that improve coverage implicitly regularizes the hypothesis class $\mathcal{H}$, helping to prevent overfitting and enhance generalization.

**Confidence Term**    This term captures the statistical reliability of empirical estimates and decreases uniformly with the number of labeled samples $m$. In its standard form, it is sample-agnostic, depending only on the sample size rather than which instances are labeled. However, this characterization holds most directly under i.i.d. assumptions. In AL settings, additional conditions are required to establish analogous bounds. As such, while this term does not directly inform sample selection, it underscores a key insight: acquiring any additional labeled point contributes to tightening the generalization guarantee. When $m$ is small, the confidence term dominates the bound, highlighting the value of rapid labeling even of moderately informative points in the early stages of AL.

### 3.6    IMPACT OF BUDGET SIZE

The labeled budget size $m$ plays a critical role in shaping the generalization error components behavior. Each term in the bound exhibits distinct sensitivity to increases in $m$: **empirical risk** typically decreases as targeted sampling corrects high-loss errors; **distributional discrepancy** shrinks as the labeled set grows more diverse and representative of the true data distribution, particularly across classes; **model complexity** is reduced through enhanced coverage and space exploration, which impose tighter functional constraints on the hypothesis class; and **confidence** improves uniformly with $m$, independent of the specific samples selected, as it depends solely on the overall sample size.

The relative importance of the generalization error components evolves with the labeling budget. At small budgets, the generalization error is primarily driven by discrepancy and empirical risk, where sample selection has the greatest influence on model learning. As the budget grows, discrepancy saturates, and the roles of model complexity and empirical risk become more prominent. The confidence term continues to tighten the bound, but remains agnostic to sample selection. This dynamic underscores the need for adaptive strategies across budget regimes (Table 1): representation and diversity are essential in early stages, while broader coverage and uncertainty refinement become beneficial with larger budgets. The proposed framework thus provides a principled and interpretable foundation for analyzing and predicting AL behavior under varying annotation constraints.

Table 1: Key AL criteria across annotation budget regimes. This table highlights which principles are most impactful at different stages, guiding strategy design and evaluation.

| Budget Regime | Representation | Coverage | Uncertainty |
|---|---|---|---|
| Small | High | Medium | Low |
| Medium | Medium | High | Low |
| High | Low | Medium | High |

## 4 EXPERIMENTS AND RESULTS

We evaluate several state-of-the-art methods on diverse vision benchmarks, ensuring reproducibility and fair comparison by following the original hyperparameters and protocols in Hacohen et al. (2022). The methods include: **Random** as uniform sampling baseline; **TypiClust** as representation-based clustering that selects typical (high-density) and diverse samples; **Uncertainty** that selects samples with lowest maximum softmax probability; **Entropy** that targets highest predictive entropy to capture epistemic uncertainty; **Margin** that focuses on smallest difference between top class probabilities, emphasizing decision boundary ambiguity; **ProbCover** that maximizes probabilistic coverage of the feature space; and **UHerding** as a hybrid method combining uncertainty and geometric coverage to promote both informativeness and diversity.

We evaluate the proposed framework in fully supervised settings across datasets of varying complexity and budget regimes, including **CIFAR-10/100** and **ImageNet-50/100/200**. In each AL round, a fixed number of samples is queried, added to the labeled pool, and used to retrain **ResNet-18** or **ResNet-50** from scratch. ResNet-18 is trained on CIFAR-10/100 for 100 epochs, and ResNet-50 on ImageNet-50/100/200 for 200 epochs. We use SGD with Nesterov momentum (0.9), weight decay (0.0003), cosine learning rate (0.025), and batch sizes ranging from 32 to 512. Standard data augmentations such as random cropping and horizontal flipping are applied in all experiments.

In some experiments, we use embeddings from self-supervised models for sample selection in the AL pipeline. Specifically, **SimCLR** Chen et al. (2020a) is pretrained on ImageNet subsets for 500 epochs using a ResNet-50 encoder with a two-layer projection head (128-dimensional output). We extract 2048-dimensional embeddings from the penultimate layer. We also use features from pretrained models including **BYOL** Grill et al. (2020), **MoCov2+** Chen et al. (2020b), and **MoCov3** Chen et al. (2021), all based on ResNet-50. Features are standardized using the training set's mean and standard deviation, with the same normalization applied to validation and test sets prior to AL.

### 4.1 BUDGET SIZE

Figure 1 shows the behavior of AL strategies under increasing annotation budgets. At the smallest budget (B=10), **Random** sampling achieves moderate class-level coverage by chance, partially spanning most clusters. **TypiClust** ensures strong class-level representation through clustering, making it well-suited for low-budget scenarios. In contrast, uncertainty-based methods like **Uncertainty** and **UHerding** tend to focus on ambiguous or noisy samples near cluster boundaries or in densely overlapping regions, often failing to achieve broad class coverage in cold-start regimes. **ProbCover** exhibits an intermediate behavior, balancing local density and spatial coverage. While it selects samples from representative areas, it may also oversample in high-density subregions, like a localized segment of the orange cluster, leading to redundancy and reduced diversity in the selected batch.

As the budget grows (B=50 to B=500), **TypiClust** continues to prioritize dense regions but struggles to expand into uncertain or outlier areas near the centers. While this yields improved coverage, it tends to overlook critical boundary regions, essential for refining decision boundaries. By B=1000, TypiClust's selections remain confined to dense clusters, with central areas still undersampled, limiting its ability to capture difficult edge cases. In contrast, uncertainty strategies focus on more ambiguous regions but fail to achieve broad spatial coverage. **UHerding** concentrates on cluster centers, neglecting peripheral instances that may contain valuable class-discriminative information. **Random** sampling consistently provides uniform coverage across budgets. Despite its simplicity, this approach delivers robust performance by ensuring both representation and space-filling, supporting its value as a strong baseline. **ProbCover** expands its reach in representation and boundary regions but occasionally oversamples densely packed subclusters, introducing redundancy that compromises selection efficiency. Additional visualizations for ImageNet are provided in the Appendix.

### 4.2 DATASET COMPLEXITY

Figure 2 compares AL strategies on CIFAR datasets across varying annotation budgets. The results reveal a consistent trend: in the early rounds, representation-based methods outperform others due to their ability to capture broad data diversity. As the budget increases, coverage-driven strategies begin to gain an advantage by exploring the feature space more effectively. At higher budgets, uncertainty-based methods gradually catch up, as their focus on ambiguous instances becomes more

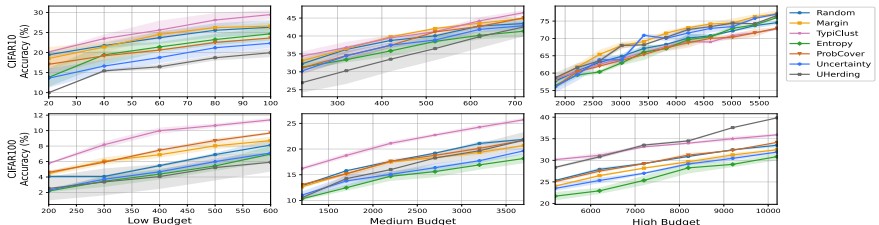

Figure 2: Performance of AL strategies across varying annotation budgets on CIFAR datasets for 3 runs. TypiClust excels early due to representation; coverage-based methods dominate mid-range budgets; uncertainty-based methods improve later. Hybrid methods perform robustly throughout.

beneficial. Hybrid approaches maintain robust performance across a wider range of budget regimes, demonstrating their adaptability. Additional results for ImageNet are provided in the Appendix.

TypiClust excels at small budgets by capturing class-level diversity. As the budget increases, performance differences narrow, with most strategies converging around B=1000. On CIFAR-100 with greater class complexity, performance trends diverge more. TypiClust remains strong early, but as the budget grows, uncertainty methods (Entropy and Margin) and UHerding outperform in mid- to high-budget regimes. UHerding, though weaker initially due to over-focusing on uncertain samples in dense regions, eventually achieves the best performance once decision boundaries form and uncertainty refinement becomes critical. This trend confirms our theoretical analysis: after representative labeling and coarse class separation informativeness becomes the key driver of generalization.

## 4.3 COLD-START PROBLEM

The cold-start phase is challenging for AL, as models begin with minimal or no labeled data. Theoretically, this stage emphasizes minimizing the distribution discrepancy term, maximizing coverage and representation in the initial selections. As shown in Figure 1, uncertainty-based methods tend to select ambiguous or difficult samples early (near decision boundaries or in visually confusing regions), resulting in limited class coverage and poor representation of the global data distribution.

Representation methods are well-suited for cold starts. By clustering the feature space and selecting typical samples, they promote broad class coverage, helping the model form an initial hypothesis. However, TypiClust's static strategy limits its utility in later rounds, as it continues sampling from high-density regions, introducing redundancy and yielding diminishing returns. To maintain generalization gains, AL strategies must evolve from early emphasis on representation to mid-stage exploration and late-stage refinement. This dynamic shift aligns with the theoretical contributions of discrepancy, complexity, and empirical risk in reducing generalization error across budgets.

### 4.3.1 SELF-SUPERVISED EMBEDDINGS

Self-supervised learning (SSL) mitigates cold-start uncertainty by structuring the feature space before labels, improving effective supervision, class separability, and boosting early accuracy. As shown in Figure 5, models using SSL features consistently achieve higher accuracy at low budgets. SSL embeddings naturally cluster inputs, reducing the need for explicit clustering or redundancy-aware strategies and implicitly promoting representation and diversity. As a result, AL methods can

Table 2: Average runtime per AL round (minutes) for small and large budgets across datasets. The ratio columns report the large-to-small budget runtime ratios. Inefficient scaling ($>2$) is highlighted.

| Method | CIFAR-10 | | Ratio | CIFAR-100 | | Ratio | ImageNet-50 | | Ratio | ImageNet-100 | | Ratio | ImageNet-200 | | Ratio |
|---|---|---|---|---|---|---|---|---|---|---|---|---|---|---|---|
| | B=10 | B=1000 | $B_{1000}/B_{10}$ | B=100 | B=1000 | $B_{1000}/B_{100}$ | B=50 | B=1000 | $B_{1000}/B_{50}$ | B=100 | B=1000 | $B_{1000}/B_{100}$ | B=200 | B=1000 | $B_{1000}/B_{200}$ |
| Random | 1.49 | 2.50 | 1.68 | 1.44 | 2.56 | 1.78 | 57.64 | 92.83 | 1.61 | 70.10 | 184.27 | **2.63** | 170.32 | 184.27 | 1.08 |
| Margin | 2.15 | 2.96 | 1.38 | 1.63 | 2.33 | 1.43 | 71.87 | 123.24 | 1.71 | 145.17 | 139.63 | 0.96 | 182.11 | 139.63 | 0.77 |
| Entropy | 2.33 | 2.48 | 1.06 | 1.54 | 2.47 | 1.60 | 46.36 | 91.04 | 1.96 | 163.53 | 186.39 | 1.14 | 163.53 | 314.39 | **1.92** |
| Uncertainty | 2.33 | 2.42 | 1.04 | 1.59 | 2.26 | 1.42 | 47.07 | 93.01 | 1.98 | 92.30 | 179.47 | 1.94 | 284.97 | 193.61 | 0.68 |
| TypiClust | 2.48 | 16.84 | **6.79** | 3.08 | 16.24 | **5.27** | 66.55 | 217.89 | **3.27** | 85.48 | 171.39 | **2.01** | 152.42 | 275.36 | 1.81 |
| ProbCover | 2.38 | 2.92 | 1.23 | 1.33 | 2.32 | 1.74 | 43.24 | 87.37 | **2.02** | 145.77 | 135.92 | 0.93 | 199.73 | 207.17 | 1.04 |
| UHerding | 1.47 | 4.07 | **2.77** | 2.01 | 2.99 | 1.49 | 56.55 | 144.26 | **2.55** | 94.15 | 170.61 | 1.81 | 226.19 | 260.99 | 1.15 |

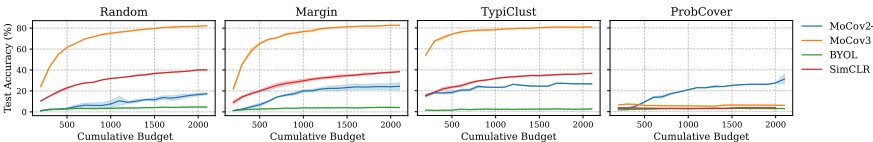

Figure 3: Low-budget performance of selected AL methods on ImageNet-100 using different embeddings. SSL enhances early-stage accuracy by providing meaningful features, particularly benefiting TypiClust. For ProbCover, the fixed-radius selection hampers training across embeddings.

replace costly techniques like $k$-means with lightweight, representation-aware sampling, improving both performance and efficiency (see Appendix for details).

## 4.4 TRAINING TIME COMPLEXITY

Scalability is critical in real-world AL, particularly with high-dimensional or large-scale datasets. Table 2 shows per-round runtimes at small and large budgets, with scalability measured as the ratio between them. Random, Margin, and Entropy exhibit consistent runtimes due to their low computational cost. TypiClust incurs high early-round cost from clustering, with runtime ratios exceeding $6\times$ on CIFAR-10. As dataset complexity grows, the number of clusters is typically capped, leading to more stable runtimes. UHerding displays a similar pattern, with added cost from neighborhood graph construction in high dimensions. Despite its conceptual complexity, ProbCover remains relatively efficient by avoiding global structures and relying on fixed-radius estimates, offering a strong balance between performance and scalability, and making it well-suited for large-scale AL.

Table 3 shows that initializing AL with SSL embeddings significantly reduces runtime. In many cases, the reduction exceeds $50\text{-}100\times$ for Margin, Entropy, and UHerding. On large-scale datasets (ImageNet-200), embedding-based models achieve substantial speedups, for example, $97\times$ for Entropy and $58\times$ for UHerding. These results highlight the scalability benefits of incorporating SSL into AL pipelines, enabling faster and more efficient sample selection in high-dimensional settings.

## 5 CONCLUSION

In this work, we introduced a generalization-theoretic framework for active learning that maps the principles of uncertainty, representation, and coverage to the components of generalization error: empirical risk, distributional discrepancy, model complexity, and confidence. This mapping provides the first unified foundation for understanding why different strategies succeed under different annotation budgets. Our experiments on CIFAR and ImageNet subsets, including settings with pretrained and self-supervised representations, reveal consistent regime-dependent patterns: representation is most valuable in early rounds to mitigate cold-start issues, coverage dominates in mid-budget regimes by promoting diversity, and uncertainty becomes most effective once decision boundaries are partially learned. While per-sample reductions in model complexity are modest, we showed that their cumulative effect across acquisition rounds is substantial. Rather than introducing a new method, our contribution is a unifying and generalizable framework that not only explains existing strategies but also serves as a blueprint for principled AL design, bridging empirical practice with theoretical understanding. Future work should extend this analysis to more challenging regimes, including class imbalance, label noise, and long-tailed distributions.

Table 3: Average runtime per AL round (minutes) on ImageNet with and without MoCov3 embeddings for budget B=1000. Efficient speedups (>1) are highlighted.

| Method | ImageNet-50 | | Speedup | ImageNet-100 | | Speedup | ImageNet-200 | | Speedup |
| | no SSL | SSL | SSL / no SSL | no SSL | SSL | SSL / no SSL | no SSL | SSL | SSL / no SSL |
|---|---|---|---|---|---|---|---|---|---|
| Random | 92.82 | 0.57 | **162.84** | 184.27 | 1.56 | **118.12** | 184.27 | 3.17 | **58.13** |
| Margin | 123.24 | 0.68 | **181.24** | 135.88 | 1.41 | **96.37** | 139.63 | 2.40 | **58.18** |
| Entropy | 91.04 | 0.63 | **144.51** | 186.39 | 1.80 | **103.55** | 314.39 | 3.22 | **97.64** |
| Uncertainty | 93.01 | 0.76 | **122.38** | 179.47 | 1.52 | **118.07** | 193.61 | 3.07 | **63.07** |
| TypiClust | 217.89 | 6.63 | **32.86** | 171.39 | 2.33 | **73.56** | 275.36 | 6.29 | **43.78** |
| ProbCover | 87.37 | 1.06 | **82.42** | 135.92 | 1.74 | **78.11** | 207.17 | 4.51 | **45.94** |
| UHerding | 144.26 | 1.20 | **120.22** | 170.61 | 2.64 | **64.62** | 260.99 | 9.22 | **28.31** |

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

# A APPENDIX

This section provides additional experimental details, along with extended visualizations and tables that further illustrate the behavior of AL methods across budget regimes. These results complement the main paper and offer deeper insights into the performance dynamics of each technique. All experiments were conducted on NVIDIA A100, A40, and RTX 6000 GPUs (48-80 GB VRAM) to ensure consistency, scalability, and reproducibility across hardware setups.

## A.1 SCALING THEOREMS

We present additional theorems and proofs related to class scaling for each of the three core components of generalization error: empirical risk, distributional discrepancy, and model complexity.

**Theorem 1 (Class-Scaling of Empirical Risk).** *For classification tasks with cross-entropy loss, the empirical risk scales logarithmically with the number of classes $C$ as $\mathcal{O}(\log C)$.*

*Proof.* The maximum empirical risk under cross-entropy loss occurs when the predicted class probabilities are uniform, i.e., $h(x) = [1/C, \ldots, 1/C]$. In this case, the loss for each sample is $-\log(1/C)$, implying that the worst-case empirical risk also scales as $\log C$. □

**Theorem 2 (Class-Scaling of Discrepancy).** *Under class-balanced sampling, the distribution discrepancy scales as $\mathcal{O}(\sqrt{C/m})$.*

*Proof.* Assume each class is equally represented in the queried set. By the Central Limit Theorem, the estimation error for the mean of a bounded function per class scales as $\mathcal{O}(1/\sqrt{m/C})$. The total discrepancy aggregates these independent class errors, scaling as $\mathcal{O}(\sqrt{C/m})$. □

**Theorem 3 (Class-Scaling of Complexity).** *For multiclass classification with $C$ classes, the Rademacher complexity of the loss-composed hypothesis class scales as $\mathcal{O}(\sqrt{\log C/m})$.*

*Proof.* Standard structural results in statistical learning theory show that the Rademacher complexity for multiclass classification with $C$ classes and linear hypotheses scales as $\mathcal{O}(\sqrt{\log C})$, with bounded inputs and a Lipschitz loss function. Averaging over $m$ labeled samples yields the bound. □

## A.2 T-SNE PLOTS

This section provides t-SNE visualizations for the ImageNet-50 dataset. As shown in Figure 4, and following the same setup as in CIFAR-10, we project the data into 2D using MoCov3 embeddings and color points according to their ground-truth class labels. The pretrained features exhibit relatively clear separation among the 50 classes, with some residual overlap and uncertainty near the center of the embedding space, though less pronounced than in CIFAR-10.

The observed patterns on ImageNet-50 align with those on CIFAR-10, supporting the generalizability of our findings. At the smallest budget (B=50), methods such as Random, TypiClust, and ProbCover (denoted as Ⓐ) effectively cover representative regions of the embedding space, with TypiClust achieving the most comprehensive class-level coverage. In contrast, uncertainty-based strategies (e.g., Entropy, Margin, UHerding, Uncertainty) often focus on ambiguous or noisy regions, resulting in redundant selections that fail to span all classes.

As the budget increases (denoted as Ⓑ), these trends persist. Uncertainty-driven methods continue targeting uncertain points, while representativeness-based approaches emphasize spatial coverage. While the latter improves early-stage performance, it may limit exploration of decision boundaries in later rounds, thereby constraining potential accuracy gains.

A notable limitation was observed in the default behavior of ProbCover (denoted as Ⓒ): a coverage radius of 0.6 was too large for ImageNet-50. In dense clusters, the selection mechanism repeatedly chose points from a single region, neglecting others. This led to insufficient class coverage and reduced labeling efficiency.

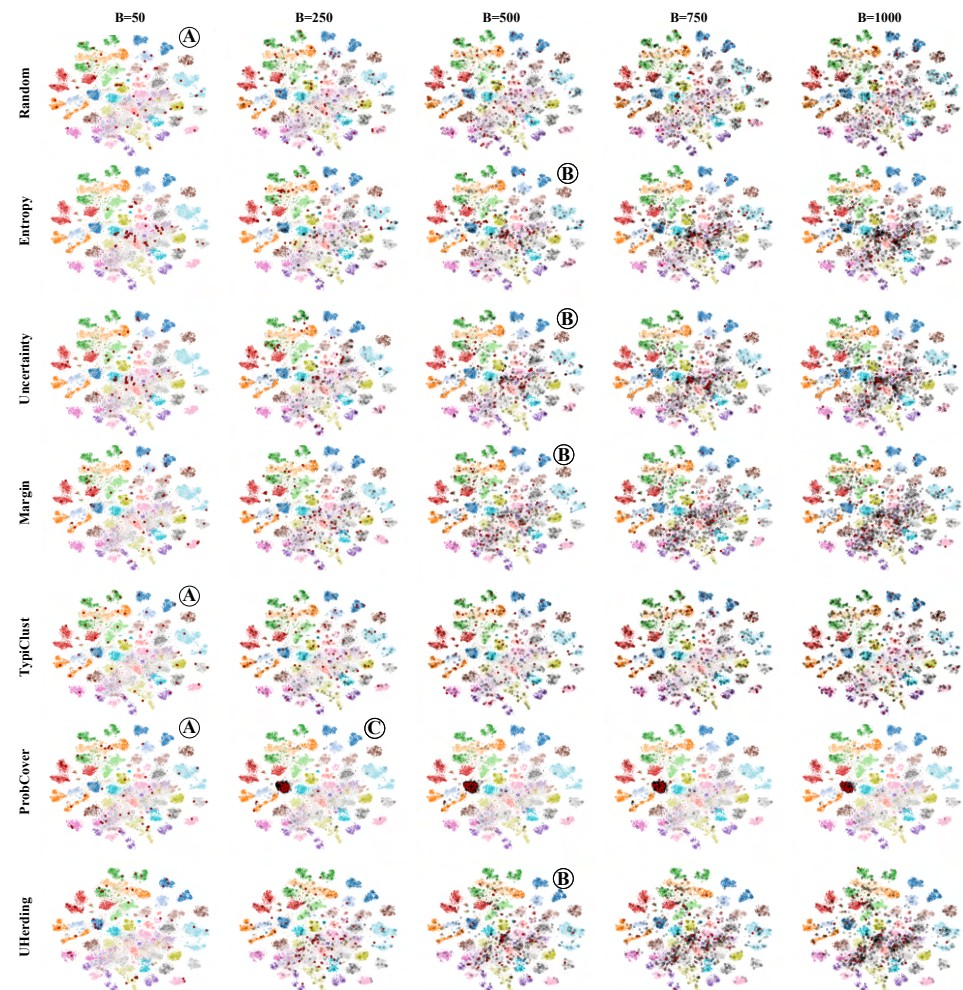

Figure 4: t-SNE visualization of sample selection behavior across different AL strategies on ImageNet-50 at increasing annotation budgets (B). Points are projected into 2D using self-supervised embeddings and colored by ground-truth class. Black points indicate previously labeled samples; red points denote newly selected samples in the current round. Representativeness-based methods (e.g., TypiClust) emphasize broad class coverage early on (See Ⓐ), while uncertainty-based methods focus on ambiguous regions. As the budget increases, strategies gradually diverge in prioritizing informativeness (See Ⓑ), representativeness, and coverage. ProbCover relying on pre-defined radius repeatedly chose points within a single region (See Ⓒ).

### A.3 DATA COMPLEXITY AND BUDGET SIZE

This section presents additional results analyzing the effects of dataset complexity and labeling budget size on AL performance, shown in Tables 4, 5, 6, 7, and 8. Across datasets of increasing complexity, consistent trends emerge: in early rounds, representativeness-based methods perform best by capturing broad data coverage. As the budget grows, diversity- and coverage-based strategies gain an advantage by reducing redundancy and expanding the labeled space. At higher budgets, uncertainty-based methods become more effective, helping resolve remaining ambiguities and refine decision boundaries.

### A.4 IMAGENET SELF-SUPERVISED EMBEDDINGS

This section presents additional results comparing self-supervised learning (SSL) embeddings on the ImageNet-50 dataset. As shown in Figure 5, MoCov3 consistently yields the best performance

Table 4: Classification accuracy (mean±SD) of various AL methods on CIFAR-10 across three runs with increasing annotation budgets. All models are trained from scratch.

| Budget | Entropy | Margin | ProbCover | Random | UHerding | Uncertainty | TypiClust |
|---|---|---|---|---|---|---|---|
| 20 | 13.82 ± 2.34 | 18.53 ± 2.86 | 17.07 ± 2.75 | 19.47 ± 0.63 | 10.00 ± 0.00 | 13.61 ± 2.39 | 20.17 ± 0.09 |
| 40 | 19.45 ± 2.59 | 21.41 ± 1.32 | 19.19 ± 1.18 | 21.75 ± 0.66 | 15.44 ± 0.31 | 16.64 ± 1.56 | 23.49 ± 1.05 |
| 60 | 21.41 ± 1.25 | 24.55 ± 0.88 | 20.63 ± 0.19 | 23.72 ± 1.54 | 16.46 ± 0.68 | 18.76 ± 1.41 | 25.63 ± 2.36 |
| 80 | 23.25 ± 1.40 | 26.23 ± 0.48 | 22.53 ± 0.13 | 25.57 ± 1.77 | 18.71 ± 0.70 | 21.21 ± 1.72 | 28.12 ± 1.76 |
| 100 | 24.69 ± 1.69 | 26.62 ± 0.90 | 23.65 ± 0.00 | 26.33 ± 1.77 | 19.97 ± 1.10 | 22.33 ± 1.13 | 29.49 ± 1.10 |
| 220 | 31.27 ± 0.92 | 33.06 ± 1.17 | 31.06 ± 1.79 | 32.24 ± 1.61 | 26.98 ± 2.79 | 30.23 ± 0.67 | 34.39 ± 1.29 |
| 320 | 33.39 ± 0.77 | 36.43 ± 0.51 | 34.53 ± 1.27 | 36.25 ± 0.93 | 30.30 ± 3.69 | 34.45 ± 1.33 | 36.77 ± 1.61 |
| 420 | 35.84 ± 1.53 | 39.78 ± 0.42 | 37.39 ± 0.81 | 38.74 ± 0.76 | 33.53 ± 3.40 | 37.43 ± 0.71 | 39.47 ± 0.78 |
| 520 | 38.49 ± 1.72 | 42.08 ± 0.21 | 41.13 ± 1.24 | 40.04 ± 0.36 | 36.48 ± 3.28 | 38.87 ± 0.89 | 41.18 ± 0.21 |
| 620 | 40.17 ± 1.11 | 43.57 ± 0.61 | 42.64 ± 0.05 | 42.80 ± 0.61 | 39.80 ± 3.18 | 41.83 ± 0.83 | 44.21 ± 1.05 |
| 720 | 41.33 ± 1.62 | 44.90 ± 0.31 | 45.05 ± 0.30 | 43.47 ± 1.01 | 42.52 ± 2.63 | 42.87 ± 1.20 | 46.51 ± 0.69 |
| 1820 | 56.44 ± 1.08 | 58.17 ± 0.72 | 56.09 ± 0.50 | 57.69 ± 0.42 | 58.66 ± 1.37 | 56.05 ± 1.69 | 58.35 ± 0.42 |
| 2220 | 59.45 ± 0.71 | 61.75 ± 0.90 | 60.37 ± 0.53 | 60.62 ± 0.78 | 61.62 ± 1.28 | 59.58 ± 1.79 | 60.45 ± 0.48 |
| 2620 | 60.35 ± 0.60 | 65.40 ± 1.30 | 62.01 ± 0.41 | 63.09 ± 0.66 | 63.54 ± 0.41 | 63.86 ± 1.63 | 62.63 ± 0.51 |
| 3020 | 62.95 ± 0.68 | 67.96 ± 0.97 | 63.70 ± 1.20 | 64.86 ± 0.78 | 67.96 ± 0.84 | 63.81 ± 0.90 | 64.10 ± 0.73 |
| 3420 | 66.03 ± 2.27 | 69.09 ± 0.77 | 65.35 ± 0.33 | 67.09 ± 0.85 | 68.15 ± 0.29 | 70.89 ± 0.43 | 66.06 ± 0.60 |
| 3820 | 67.09 ± 0.61 | 71.51 ± 0.66 | 67.29 ± 1.01 | 68.29 ± 0.17 | 70.32 ± 0.25 | 70.03 ± 0.94 | 67.66 ± 0.69 |
| 4220 | 69.66 ± 1.60 | 73.11 ± 0.50 | 68.76 ± 1.43 | 70.10 ± 1.16 | 72.56 ± 0.59 | 71.57 ± 1.53 | 69.10 ± 0.31 |
| 4620 | 70.48 ± 0.57 | 74.17 ± 0.39 | 70.02 ± 0.56 | 70.77 ± 0.54 | 73.40 ± 0.04 | 72.89 ± 1.19 | 68.97 ± 0.46 |
| 5020 | 72.93 ± 0.65 | 74.45 ± 1.33 | 70.36 ± 0.82 | 72.16 ± 0.19 | 74.41 ± 0.63 | 73.41 ± 0.80 | 70.83 ± 0.87 |
| 5420 | 73.90 ± 0.48 | 76.25 ± 0.71 | 71.60 ± 1.17 | 73.69 ± 0.82 | 74.24 ± 0.16 | 75.80 ± 0.27 | 71.63 ± 0.28 |
| 5820 | 76.03 ± 0.26 | 77.16 ± 0.99 | 72.95 ± 0.19 | 74.56 ± 0.56 | 76.61 ± 0.32 | 77.01 ± 0.35 | 72.71 ± 0.25 |

Table 5: Classification accuracy (mean±SD) of various AL methods on CIFAR-100 across three runs with increasing annotation budgets. All models are trained from scratch.

| Budget | Entropy | Margin | ProbCover | Random | UHerding | Uncertainty | TypiClust |
|---|---|---|---|---|---|---|---|
| 200 | 2.19 ± 0.15 | 4.44 ± 0.28 | 4.60 ± 0.25 | 4.07 ± 0.22 | 2.52 ± 1.54 | 2.31 ± 0.22 | 5.76 ± 0.11 |
| 300 | 3.45 ± 0.74 | 6.07 ± 0.50 | 5.91 ± 0.08 | 4.07 ± 0.15 | 3.36 ± 1.75 | 3.71 ± 0.30 | 8.18 ± 0.42 |
| 400 | 4.37 ± 0.70 | 6.88 ± 0.40 | 7.47 ± 0.18 | 5.47 ± 0.16 | 4.06 ± 1.60 | 4.67 ± 0.29 | 9.99 ± 0.41 |
| 500 | 5.43 ± 0.64 | 8.03 ± 0.43 | 8.71 ± 0.28 | 6.90 ± 0.23 | 5.22 ± 1.67 | 5.99 ± 0.41 | 10.66 ± 0.27 |
| 600 | 6.97 ± 1.06 | 8.71 ± 0.51 | 9.69 ± 0.13 | 8.13 ± 0.73 | 5.91 ± 1.21 | 7.10 ± 0.34 | 11.39 ± 0.26 |
| 1200 | 10.31 ± 0.30 | 12.64 ± 0.28 | 13.08 ± 0.13 | 12.86 ± 0.17 | 10.53 ± 0.23 | 11.03 ± 0.72 | 16.20 ± 0.37 |
| 1700 | 12.47 ± 0.60 | 15.20 ± 0.27 | 15.18 ± 0.56 | 15.78 ± 0.03 | 14.23 ± 0.72 | 13.79 ± 1.06 | 18.76 ± 0.47 |
| 2200 | 14.71 ± 0.92 | 17.50 ± 0.07 | 17.62 ± 0.45 | 17.60 ± 0.03 | 16.00 ± 1.27 | 15.07 ± 0.55 | 21.12 ± 0.44 |
| 2700 | 15.61 ± 1.12 | 18.32 ± 0.21 | 18.77 ± 0.27 | 19.20 ± 0.06 | 18.26 ± 1.20 | 16.37 ± 0.84 | 22.73 ± 0.33 |
| 3200 | 16.92 ± 0.93 | 19.35 ± 0.48 | 20.13 ± 0.33 | 21.09 ± 0.41 | 19.67 ± 1.44 | 17.73 ± 0.50 | 24.26 ± 0.45 |
| 3700 | 18.16 ± 0.93 | 20.69 ± 0.10 | 21.71 ± 0.25 | 21.89 ± 0.07 | 21.74 ± 1.56 | 19.65 ± 0.67 | 25.71 ± 0.36 |
| 5200 | 21.69 ± 0.97 | 24.03 ± 0.16 | 25.08 ± 0.33 | 25.33 ± 0.12 | 28.37 ± 0.24 | 23.49 ± 0.61 | 30.16 ± 0.29 |
| 6200 | 22.93 ± 0.74 | 26.45 ± 0.32 | 27.50 ± 0.41 | 27.90 ± 0.42 | 30.81 ± 0.42 | 25.33 ± 0.67 | 31.16 ± 0.59 |
| 7200 | 25.34 ± 1.10 | 28.19 ± 0.12 | 29.21 ± 0.07 | 29.25 ± 0.18 | 33.53 ± 0.26 | 26.99 ± 0.26 | 33.11 ± 0.16 |
| 8200 | 28.25 ± 1.31 | 29.66 ± 0.46 | 31.21 ± 0.55 | 30.90 ± 0.03 | 34.50 ± 0.24 | 29.15 ± 0.33 | 33.97 ± 0.27 |
| 9200 | 29.06 ± 0.65 | 31.25 ± 0.50 | 32.32 ± 0.00 | 32.44 ± 0.14 | 37.58 ± 0.00 | 30.46 ± 0.57 | 35.01 ± 0.54 |
| 10200 | 30.86 ± 0.88 | 32.45 ± 0.27 | 34.16 ± 0.00 | 33.49 ± 0.52 | 39.84 ± 0.00 | 31.91 ± 0.22 | 35.91 ± 0.31 |

across most AL methods and datasets, likely due to its advanced architecture and training scheme. MoCov2 and SimCLR also provide notable gains, while BYOL generally lags behind.

MoCov3, in particular, significantly boosts early-stage performance for many AL methods. For instance, TypiClust surpasses 60% accuracy after just the first labeling round. More broadly, SSL embeddings accelerate training and improve sample efficiency, achieving accuracies that previously required labeling several percent of the dataset with under 1% labeled data.

Table 6: Classification accuracy (mean±SD) of various AL methods on ImageNet-50 across three runs with increasing annotation budgets. All models are trained from scratch.

| Budget | Entropy | Margin | ProbCover | Random | UHerding | Uncertainty | TypiClust |
|--------|---------|--------|-----------|--------|----------|-------------|-----------|
| 200 | 3.0 ± 0.5 | 5.0 ± 0.8 | 7.6 ± 0.1 | 8.0 ± 0.4 | 3.3 ± 0.8 | 4.5 ± 0.1 | 10.88 ± 0.8 |
| 400 | 5.6 ± 0.7 | 10.5 ± 0.3 | 9.8 ± 0.7 | 11.2 ± 0.2 | 6.5 ± 0.8 | 10.6 ± 1.4 | 18.24 ± 0.7 |
| 600 | 8.3 ± 1.4 | 15.5 ± 0.9 | 10.3 ± 0.5 | 15.6 ± 0.5 | 7.8 ± 3.0 | 13.5 ± 0.4 | 23.08 ± 0.1 |
| 800 | 11.3 ± 3.7 | 19.4 ± 2.1 | 9.2 ± 0.2 | 20.1 ± 0.7 | 9.9 ± 0.9 | 18.0 ± 0.6 | 26.78 ± 1.3 |
| 1000 | 14.7 ± 1.7 | 23.2 ± 0.3 | 9.4 ± 1.0 | 22.7 ± 0.7 | 13.1 ± 0.2 | 19.2 ± 0.6 | 32.16 ± 1.2 |
| 1400 | 25.9 ± 1.6 | 26.4 ± 3.0 | 26.3 ± 0.2 | 26.6 ± 1.1 | 26.4 ± 1.2 | 26.3 ± 1.6 | 33.88 ± 1.5 |
| 1800 | 30.3 ± 0.7 | 29.1 ± 3.5 | 34.1 ± 0.3 | 32.3 ± 0.8 | 30.2 ± 1.6 | 30.2 ± 0.5 | 35.76 ± 0.9 |
| 2200 | 38.4 ± 0.9 | 36.8 ± 0.7 | 34.7 ± 1.4 | 35.8 ± 1.6 | 35.5 ± 3.4 | 35.0 ± 0.5 | 34.68 ± 1.1 |
| 2600 | 38.3 ± 5.0 | 41.1 ± 1.5 | 35.3 ± 0.1 | 43.3 ± 1.7 | 42.5 ± 2.1 | 41.8 ± 0.7 | 39.12 ± 0.7 |
| 3000 | 45.3 ± 3.5 | 45.0 ± 2.4 | 37.4 ± 0.1 | 45.5 ± 1.1 | 48.2 ± 0.7 | 44.5 ± 4.2 | 41.04 ± 0.4 |
| 4000 | 51.5 ± 0.2 | 52.9 ± 1.0 | 55.9 ± 2.2 | 52.7 ± 0.6 | 54.7 ± 0.6 | 51.9 ± 0.5 | 43.88 ± 0.5 |
| 5000 | 59.1 ± 0.3 | 60.0 ± 0.4 | 59.6 ± 3.3 | 56.7 ± 0.2 | 58.9 ± 3.6 | 57.0 ± 0.5 | 49.56 ± 0.3 |
| 6000 | 61.6 ± 1.4 | 63.3 ± 0.3 | 63.2 ± 0.8 | 60.1 ± 0.9 | 66.0 ± 1.8 | 61.9 ± 0.2 | 54.20 ± 0.4 |
| 7000 | 62.8 ± 2.7 | 67.9 ± 0.3 | 65.6 ± 0.8 | 67.1 ± 0.4 | 69.6 ± 1.3 | 66.1 ± 2.3 | 58.12 ± 0.8 |

Table 7: Classification accuracy (mean±SD) of various AL methods on ImageNet-100 across three runs with increasing annotation budgets. All models are trained from scratch.

| Budget | Entropy | Margin | ProbCover | Random | UHerding | Uncertainty | TypiClust |
|--------|---------|--------|-----------|--------|----------|-------------|-----------|
| 200 | 3.04 ± 0.52 | 5.04 ± 0.84 | 7.58 ± 0.06 | 7.96 ± 0.40 | 3.32 ± 0.76 | 4.54 ± 0.02 | 5.01 ± 0.59 |
| 400 | 5.62 ± 0.74 | 10.52 ± 0.32 | 9.78 ± 0.70 | 11.24 ± 0.20 | 6.50 ± 0.78 | 10.60 ± 1.44 | 7.28 ± 0.76 |
| 600 | 8.26 ± 1.38 | 15.46 ± 0.90 | 10.32 ± 0.48 | 15.58 ± 0.54 | 7.84 ± 3.00 | 13.46 ± 0.38 | 10.36 ± 1.61 |
| 800 | 11.32 ± 3.72 | 19.40 ± 2.08 | 9.20 ± 0.20 | 20.08 ± 0.72 | 9.90 ± 0.90 | 17.98 ± 0.58 | 12.72 ± 1.01 |
| 1000 | 14.66 ± 1.74 | 23.24 ± 0.32 | 9.36 ± 1.00 | 22.66 ± 0.70 | 13.12 ± 0.24 | 19.18 ± 0.58 | 15.22 ± 2.68 |
| 1400 | 25.94 ± 1.58 | 26.38 ± 2.98 | 26.34 ± 0.18 | 26.58 ± 1.06 | 26.40 ± 1.24 | 26.28 ± 1.64 | 19.40 ± 1.93 |
| 1800 | 30.28 ± 0.68 | 29.06 ± 3.46 | 34.10 ± 0.34 | 32.28 ± 0.80 | 30.22 ± 1.62 | 30.18 ± 0.50 | 23.35 ± 2.71 |
| 2200 | 38.38 ± 0.90 | 36.80 ± 0.68 | 34.68 ± 1.36 | 35.84 ± 1.56 | 35.48 ± 3.44 | 35.04 ± 0.52 | 29.16 ± 5.55 |
| 2600 | 38.26 ± 4.98 | 41.14 ± 1.50 | 35.26 ± 0.06 | 43.30 ± 1.74 | 42.50 ± 2.06 | 41.82 ± 0.74 | 31.40 ± 3.19 |
| 3000 | 45.26 ± 3.50 | 45.00 ± 2.40 | 37.42 ± 0.06 | 45.54 ± 1.10 | 48.24 ± 0.68 | 44.52 ± 4.16 | 37.25 ± 0.63 |
| 4000 | 51.54 ± 0.18 | 52.86 ± 0.98 | 57.92 ± 2.16 | 52.70 ± 0.62 | 54.66 ± 0.58 | 51.94 ± 0.46 | 44.85 ± 0.19 |
| 5000 | 59.06 ± 0.26 | 59.96 ± 0.36 | 59.62 ± 3.34 | 56.68 ± 0.20 | 61.02 ± 1.46 | 57.02 ± 0.54 | 48.58 ± 1.44 |
| 6000 | 61.58 ± 1.42 | 63.28 ± 0.32 | 63.18 ± 0.78 | 60.06 ± 0.94 | 66.00 ± 1.76 | 61.94 ± 0.18 | 52.25 ± 1.75 |
| 7000 | 62.84 ± 2.72 | 67.86 ± 0.34 | 65.60 ± 0.76 | 67.12 ± 0.04 | 69.64 ± 1.28 | 66.08 ± 2.28 | 55.64 ± 0.09 |

ProbCover requires special attention. As discussed earlier, it relies on a fixed radius and local density estimates. In highly clustered and class-separated embedding spaces (e.g., Figure 4), ProbCover tends to oversample dense regions while neglecting underrepresented ones. This limitation highlights challenges for purely density-based strategies in structured feature spaces.

Interestingly, the effectiveness of MoCov2 varies across methods. While it enables strong performance for Random, Entropy, TypiClust, and Margin, it underperforms with Uncertainty and UHerding. For ProbCover, where most embeddings perform poorly, MoCov2 uniquely yields a steady accuracy increase, suggesting better alignment with its fixed-radius criterion.

These findings highlight the value of SSL embeddings, both for feature extraction and initialization, in enhancing AL efficiency and performance, especially as dataset complexity increases.

Table 8: Classification accuracy (mean±SD) of various AL methods on ImageNet-200 across three runs with increasing annotation budgets. All models are trained from scratch.

| Budget | Entropy | Margin | ProbCover | Random | UHerding | Uncertainty | TypiClust |
|---|---|---|---|---|---|---|---|
| 200 | 1.90 ± 0.34 | 0.66 ± 0.16 | 1.69 ± 0.01 | 1.52 ± 0.05 | 0.77 ± 0.05 | 0.70 ± 0.09 | 2.82 ± 0.18 |
| 400 | 1.98 ± 0.50 | 2.05 ± 0.01 | 2.18 ± 0.06 | 1.93 ± 0.06 | 1.15 ± 0.04 | 1.77 ± 0.01 | 3.26 ± 0.64 |
| 600 | 2.66 ± 0.09 | 2.77 ± 0.18 | 2.13 ± 0.00 | 2.74 ± 0.10 | 2.49 ± 0.12 | 2.44 ± 0.17 | 4.44 ± 0.58 |
| 800 | 3.58 ± 0.28 | 4.32 ± 0.41 | 2.16 ± 0.11 | 2.90 ± 0.40 | 3.13 ± 0.45 | 3.49 ± 0.20 | 5.62 ± 1.01 |
| 1000 | 4.71 ± 0.29 | 5.89 ± 0.39 | 2.04 ± 0.14 | 4.18 ± 0.41 | 4.01 ± 0.16 | 4.29 ± 0.35 | 5.14 ± 0.49 |
| 1400 | 5.55 ± 0.42 | 8.49 ± 1.22 | 6.03 ± 0.74 | 5.66 ± 0.57 | 5.97 ± 0.77 | 5.39 ± 0.75 | 5.00 ± 0.56 |
| 1800 | 6.86 ± 1.17 | 7.89 ± 0.95 | 8.11 ± 0.14 | 8.29 ± 0.13 | 8.47 ± 0.37 | 8.01 ± 0.25 | 8.19 ± 0.67 |
| 2200 | 9.37 ± 0.20 | 10.15 ± 0.48 | 9.45 ± 0.10 | 10.70 ± 0.13 | 10.05 ± 0.83 | 10.02 ± 0.32 | 11.38 ± 0.18 |
| 2600 | 10.61 ± 0.29 | 13.41 ± 1.42 | 11.42 ± 0.62 | 10.97 ± 0.29 | 13.11 ± 0.18 | 9.95 ± 1.35 | 15.36 ± 0.35 |
| 3000 | 11.62 ± 0.68 | 19.58 ± 1.08 | 21.27 ± 1.14 | 13.36 ± 0.63 | 15.90 ± 0.18 | 14.34 ± 0.38 | 14.38 ± 1.98 |
| 4000 | 19.04 ± 0.24 | 25.06 ± 0.10 | 24.90 ± 0.22 | 19.96 ± 1.53 | 21.14 ± 0.42 | 18.82 ± 1.03 | 20.25 ± 1.16 |
| 5000 | 23.54 ± 1.01 | 28.56 ± 4.08 | 25.75 ± 0.52 | 26.01 ± 1.43 | 25.28 ± 0.54 | 24.65 ± 0.61 | 25.64 ± 3.79 |
| 6000 | 28.68 ± 0.53 | 31.79 ± 5.80 | 26.52 ± 1.28 | 29.10 ± 3.66 | 29.46 ± 0.66 | 28.74 ± 0.52 | 29.96 ± 2.27 |
| 7000 | 31.14 ± 2.68 | 34.68 ± 2.53 | 26.82 ± 1.59 | 34.20 ± 3.80 | 33.28 ± 0.56 | 34.82 ± 0.03 | 35.67 ± 3.01 |

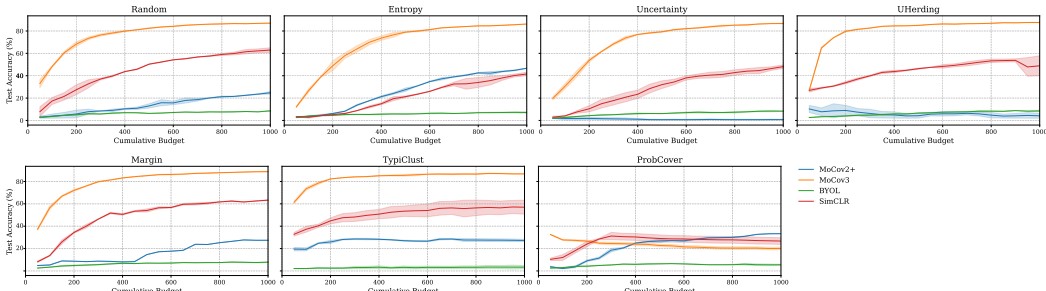

Figure 5: Low-budget performance of different AL methods on ImageNet-50 using SSL embeddings. Benefiting from semantically meaningful features, SSL boosts early-stage accuracy, especially for TypiClust. For ProbCover, fixed-radius selection hinders training with most embeddings.

