# OpenReview forum: "When Uncertainty, Coverage, and Representation Matter in Active Learning Frameworks"
_ICLR.cc/2026/Conference — Submitted to ICLR 2026_

### Official Review · Reviewer_CEPw · 2025-10-26

**Soundness:** 1
**Presentation:** 3
**Contribution:** 1
**Rating:** 2
**Confidence:** 4

**Summary:**

The paper proposes a PAC learning-based theoretical framework for analyzing which active learning strategy is most effective in different sample size regimes.

**Strengths:**

- The research question the paper focuses on is an exciting one: what sampling method works best for different data size regimes?
- The paper presents extensive experiments.

**Weaknesses:**

**Significance of the theoretical results.** It is unclear why the framework of risk bounds for domain adaptation is appropriate for capturing the benefits of active learning
- Assuming the distributional setup proposed in the paper, the upper bound of Ben-David et al seems unsuitable for characterizing the risk of active learning. To illustrate this, let us assume a noiseless and realizable learning problem. Therefore, the minimizer of the empirical risk on $S$ can always have $0$ training error. Passive learning would lead to an upper bound in equation 11 with a vanishing discrepancy term. It is a well-known fact that an uncertainty-based sampling strategy can lead to exponential improvements in sample complexity in the large sample regime. However, such a strategy would lead to a distribution $P_Q$ concentrated close to the decision boundary, which in turn would lead to a large distribution discrepancy term. Therefore, the upper bound for the AL strategy is larger than the upper bound for passive learning.
- Lines 256-258 claim that selecting samples with high loss leads to low error on the target distribution $P_Z$. This statement seems incorrect: the empirical risk term present in the left-hand side of the bound should be minimized in order to reduce the target risk, and hence, it suggests selecting samples with low loss, contrary to what uncertainty-based AL algorithms do.
- The framework proposed in section 3 lacks lower bounds, which makes it impossible to draw conclusions about how to rank AL strategies in different sample size regimes.

**Positioning in the literature.** There are a few claims in the paper that could benefit from being better placed in the context of prior works:
- There are a number of works that discuss different sample size regimes and what optimal sampling algorithms perform better, e.g. https://arxiv.org/abs/1911.09162, https://arxiv.org/pdf/2212.00772. In particular, the latter also proposes theoretical arguments for why different strategies work better in different sample size regimes.
- In addition to the works referenced in lines 105-109, there are a few others that propose sampling algorithms similar to uncertainty sampling which would be worth mentioning, such as https://www.cs.cmu.edu/~ninamf/papers/active-ls.pdf, https://arxiv.org/abs/1802.09841 etc. Notably, in this context “uncertainty sampling” can be a bit of a misnomer – uncertainty implies a notion of uncertainty which may or may not be implied by some of these methods (for instance margin-based sampling does not require that such a notion of uncertainty be assumed)


**Minor issues**
- figure 1 not particularly clear, both visually and in terms of its takeaway – while UHerding exhibits a clearly different sampling pattern at large budgets, the other methods seem largely similar
- the empirical risk in equation 11 is defined later in equation 12 as the empirical mean of the cross-entropy loss; however, equation 11 is proved in Ben-David et al to hold for the 0-1 loss.

**Questions:**

- what is the fundamental difference between representation-based and coverage-based sampling? from their definitions in section 2 they seem to target similar objectives

---

> ### Author Response · Authors · 2025-12-01
>
> We sincerely thank the reviewer for the careful reading and for the valuable comments regarding the theoretical setup and positioning. Below, we address each point and clarify some unclear aspects.
>
> (Q1) Representation/coverage-based sampling: The term coverage is overloaded in the literature, as it may refer to:
> (a) representation coverage: density or typicality in feature space
> (b) geometric coverage: spatial dispersion of selected points
> (c) probabilistic coverage: mass coverage under the data distribution
> In our framework, Representation corresponds to density-based selection, choosing points that lie in typical or high-density regions of the feature space. Coverage, in contrast, refers to geometric expansion of the labeled set, selecting samples that increase the spatial support of labeled points and thereby reduce the discrepancy between the labeled-set distribution and the underlying data distribution. This distinction allows us to separate “sampling near the center of mass” (representation) from “sampling to expand support” (coverage), which aligns with how these principles manifest in AL practice.
>
> (1) Significance of the theoretical results: We thank the reviewer for raising the concerns.
> (a) It is true that, in passive learning, minimizing the empirical risk on a noiseless dataset can yield zero training error. In active learning, however, the learner chooses which samples to label, which breaks the i.i.d. assumption. This introduces three effects that do not exist in passive learning: sampling bias (selected points are not fully representative of the underlying distribution), distributional mismatch (the labeled set and unlabeled pool lie on different empirical distributions), and acquisition-induced shift (different AL strategies impose different biases on which regions of the space are explored). Because of these effects, the discrepancy term does not vanish under AL, even when the classifier has zero empirical error on the labeled set. Therefore, discrepancy is a fundamental component of the AL decomposition, and this is why uncertainty-based sampling often underperforms in the very small-budget regime: selecting near-boundary points early tends to increase the mismatch between ​ labeled distribution PQ and ​ true data distribution  PX.
> (b) Our claim in the paper is not that selecting high-loss points minimizes the bound in general. Instead, we discuss that early in AL, when the hypothesis class is large and the representation is unstable, high-loss/uncertain points do not reduce true error effectively; meanwhile, they amplify the discrepancy term. Additionally, at larger budgets, once coverage and representation are sufficiently improved, uncertainty becomes beneficial because it reduces the hypothesis space further without substantially harming distribution match. These explain the empirical phase transition: coverage and representativeness dominate early/mid stages, while uncertainty becomes effective only after the discrepancy term has stabilized.
> (c) We acknowledge that the paper does not provide lower bounds. Our goal is not to propose a new theoretical guarantee, but to use the existing decomposition to interpret empirical AL behavior.
>
> (2) Positioning in the literature: We thank the reviewer for pointing out additional relevant works. The suggested studies are quite relevant and worth including in the related-work section. Moreover, it is important to note that most literature studies start their analysis from a budget with like 1000 labeled samples, while our experiments start with the minimum possible amount of samples (number of classes in the set) to step by step showcase the changes in the sampling selection, even in extremely small budget regimes. This is crucial as multiple available datasets for real-world use cases require annotation that is time-consuming and costly.
>
> (3) Minor issues: We agree that the first figure may not be particularly clear. For this reason, we have provided a figure for ImageNet50 in the appendix with more explanation to clarify the differences between sampling methods. Additionally, we agree with the reviewer that the original Ben-David generalization bound is stated for the 0-1 loss. However, the decomposition we use follows the surrogate-loss extension formalized in Theorem 7 of Bartlett & Mendelson (2002), which provides a generalization bound for any Lipschitz loss φ that upper-bounds the 0-1 loss. When expressed as a function of the margin, the cross-entropy loss satisfies both required conditions: (i) φ(α) ≥ 1(α ≤ 0) and (ii) φ is Lipschitz on bounded logits. Therefore, using cross-entropy in Eq. 12 is theoretically justified as an instance of the general surrogate-loss bound permitted by Theorem 7.

---

### Official Review · Reviewer_hnNz · 2025-10-31

**Soundness:** 2
**Presentation:** 4
**Contribution:** 1
**Rating:** 2
**Confidence:** 3

**Summary:**

The paper investigates active learning within a PAC learning framework, focusing on a bound that decomposes the generalization error (or true risk) into four distinct components: empirical risk, distributional discrepancy, model complexity, and confidence. The authors map the core principles of various AL strategies to these four theoretical components, offering a discussion on what each strategy effectively optimizes within the decomposition. The experimental evaluation utilizes image classification datasets and explores known AL phenomena, such as the impact of budget size or the cold-start problem.

**Strengths:**

- The paper is extremely well-written and easy to follow.
- The problem addressed is highly important, as it provides theoretical insights into what AL strategies focus on through the lens of generalization.
- The experiments cover a broad range of complexities by including CIFAR-10 and CIFAR-100, as well as different variants of ImageNet.

**Weaknesses:**

**Clarity of Theoretical Contribution**

My primary concern relates to the novelty and scope of the theoretical contribution. The paper introduces the decomposition (Eq. 11) and states, "We decompose the error into four components." However, the subsequent citation to Menden et al. suggests this decomposition may stem from related work, which leads to ambiguity, especially given that **no derivation** of the bound is provided.

- **Contribution Clarity:** It is critical to clarify whether the decomposition of the bound itself is a novel contribution or if it originates from Menden et al. If the latter, this should be explicitly stated and highlighted in the introduction to ensure precision in defining the paper's scope.
- **Framework Definition:** Given that the main body seems to be a discussion connecting established Active Learning (AL) strategies to specific terms in an existing bound, the claim of providing a "framework" needs clarification. Could you please elaborate on what constitutes the novel framework proposed here? Is it the novel combination, the specific interpretation, or something else entirely?
- **Significance:** If the decomposition is not novel and the main body primarily consists of a discussion, the overall contribution of the paper is significantly diminished, which impacts my current score. Sharpening the wording regarding the origin of the decomposition is necessary to resolve this critical point.

**Experimental Rigor and Reproducibility**

The presentation of the experimental methodology and results requires greater detail and rigor.

- **Replication and Statistics:** The absence of statistical measures (such as standard error or deviation) is a significant limitation. Given the known high variability of AL experiments [1, 2], especially at low sample sizes, a single AL run is generally insufficient to draw reliable conclusions. Did the authors repeat experiments and average metrics, or was each experiment only a single AL run? This crucial information should be provided and discussed.
- **Experimental Protocol:** While the authors mention following the protocol of Hacohen et al. (2022), this is insufficient detail. More comprehensive information regarding the experimental setup should be available, at least in the appendix, to ensure reproducibility.

**Evaluation Scope and Connection to Theory**

The evaluation section is limited in scope and lacks a clear connection to the theoretical decomposition.

- **Limited Evaluation Focus:** The evaluation primarily focuses on concepts already established in the literature (e.g., exploring early and refining late). This results in a repetition of known insights.
- **Missing Theory-Experiment Link:** Despite occasional mentions of the decomposition, the experimental design does not appear to be clearly structured to systematically study the influence of the individual components within the bound. A clearer connection demonstrating how each experiment quantitatively isolates or studies a specific component of the decomposition would substantially strengthen the contribution.
- **Qualitative Evidence:** Reliance on qualitative assessments, such as single-run t-SNE visualizations, is insufficient for drawing strong conclusions in highly variable AL settings. Quantitative metrics should be prioritized.

**Related Work and Vague formulation:**

- **Related Work:** The related work section is too limited. The discussion should be broadened to include works that have empirically investigated the influence of the specific factors (e.g., studies on uncertainty sampling [3, 4]). Discussing how the proposed theoretical perspective adds to or complements these existing empirical findings would significantly improve the paper's context and relevance.
- **Conceptual Distinction:** The distinction between "coverage" and "representativeness" in Section 4.2 remains vague. The text states that "representation-based methods capture broad data diversity" and "coverage-driven strategies... explor[e] the feature space more effectively." To an active learning researcher, both descriptions essentially imply exploring the feature space. Please provide a clear, precise, and formalized distinction between these two concepts.

**Minor Issues:**

- Fix citations to properly use parentheses.
- Improve the DPI quality of TSNE Visualizations.

[1] P. Munjal et al., “Towards Robust and Reproducible Active Learning Using Neural Networks", in *CVPR*, 2022.

[2] T. Werner et al. "A cross-domain benchmark for active learning", in *NeurIPS,* 2024.

[3] J. Li et al. "Bal: Balancing diversity and novelty for active learning." *in TPAMI* 2023.

[4] D. Huseljic et al. “The Interplay of Uncertainty Modeling and Deep Active Learning: An Empirical Analysis in Image Classification”, in *TMLR*, 2024.

**Questions:**

Could you comment on the points mentioned under “Weaknesses”?

---

> ### Author Response · Authors · 2025-12-01
>
> We sincerely thank the reviewer for the detailed, thorough, and constructive evaluation. We address each of the raised points below.
>
> (1) Contribution clarity: Our contribution does not lie in introducing a new generalization bound. Instead, it lies in mapping an existing decomposition to the AL setting and using it to explain the empirical behavior of AL methods. Recent theoretical work, such as Menden et al. (2025), studies distribution shift in AL, but it does not connect this analysis to empirical observations or to the long-standing families of AL strategies. Many AL methods have been developed empirically and lack a theory-grounded interpretation of why they work. Our contribution bridges this gap by:
> (a) Adapting the generalization bound decomposition to the AL process.
> (b) Mapping each term to a core AL principle (uncertainty, representativeness, coverage).
> (c) Explaining the phase transitions across budgets using these four terms.
> (d) Showing how existing methods implicitly optimize different subsets of components.
> This mapping provides, for the first time, a theory-grounded explanation of empirical AL behavior and suggests a principled “recipe” for designing new AL methods rather than relying solely on heuristics or trial-and-error.
>
> (2) Framework definition: By “framework,” we refer to a systematic analytical tool for interpreting AL methods through four generalization components. This provides a diagnostic lens that predicts and explains empirical behaviors such as: representation-driven selection being effective early, coverage becoming dominant in mid-budget regimes, and uncertainty dominating when the hypothesis class stabilizes at high budgets. Understanding which term dominates in which regime offers actionable insight: it becomes possible to design AL strategies that deliberately target the right component at the right stage, potentially improving performance across all budgets.
>
> (3) Significance: The significance of our work is twofold:
> (a) Unification: We provide a single decomposition that explains when and why classical AL strategies succeed or fail, clarifying several behaviors previously understood only empirically.
> (b) Diagnosis: The framework allows systematic analysis of AL behavior across budgets,  dataset sizes, and SSL methods without proposing a new algorithm.
>
> (4) Replication: All experiments in the paper were indeed repeated 3-5 times, depending on the dataset and setting. Standard deviation is shown as translucent shading in all accuracy curves and reported numerically in the appendix.
>
> (5) Experimental protocol: We agree that providing more details on the experimental setup in the appendix will ensure reproducibility.
>
> (6) Evaluation scope and connection to theory: We agree with the reviewer that although the empirical fact “representation is strongest early” is well known, the theoretical reason behind it has not been explained in prior AL literature. Addressing this gap precisely is the primary motivation of our work; we show that representation dominates early budgets due to the interaction between empirical risk, model complexity, and early-round discrepancy, and we provide the theoretical decomposition that explains why this occurs systematically across datasets and methods. Additionally, we agree that isolating the magnitude of each generalization term would strengthen the paper. So, we are conducting new experiments to quantitatively measure each of the four components’ behavior within AL to address the missing theory-experiment link.
>
> (7) Related work: We thank the reviewer for the suggested references and agree that the extension of the discussion to integrate findings from the mentioned AL papers would significantly improve the paper’s context and relevance.
>
> (8) Conceptual distinction: The term coverage is overloaded in the literature, referring to:
> (a) representation coverage: density or typicality in feature space
> (b) geometric coverage: spatial dispersion of selected points
> (c) probabilistic coverage: mass coverage under the data distribution
> In our framework, Representation corresponds to density-based selection, choosing points that lie in typical or high-density regions of the feature space. Coverage, in contrast, refers to geometric expansion of the labeled set, selecting samples that increase the spatial support of labeled points and thereby reduce the discrepancy between the labeled-set distribution and the underlying data distribution. This distinction allows us to separate “sampling near the center of mass” (representation) from “sampling to expand support” (coverage), which aligns with how these principles manifest in AL practice.

---

### Official Review · Reviewer_jJe5 · 2025-10-31

**Soundness:** 2
**Presentation:** 2
**Contribution:** 2
**Rating:** 4
**Confidence:** 4

**Summary:**

This paper presents a framework for understanding active learning (AL) through the decomposition into four components: empirical risk, distributional discrepancy, model complexity, and confidence. The authors map these components to the three core AL principles—uncertainty, representation, and coverage—providing a unified view that explains the empirical success of diverse AL strategies under different labeling budgets. The framework is validated through experiments on CIFAR and ImageNet subsets, analyzing how various AL methods behave across budget regimes and dataset complexities.

**Strengths:**

(S1) The paper provides a theoretical lens linking active learning principles to generalization theory.
(S2) The paper helps explaining previously observed trends (e.g., when uncertainty vs. diversity dominates).
(S3) The paper is generally well structured and readable.

**Weaknesses:**

(W1) Lack of Direct Validation of the Core Theoretical Claims: A central issue of the paper is that the proposed mapping between the four generalization error components and active learning principles is never directly validated. The empirical results mainly show accuracy trends and qualitative t-SNE visualizations, but do not quantify or isolate the claimed error terms. For example, the distribution discrepancy term is discussed; the model complexity term is linked to Rademacher complexity yet never estimated or tracked during learning; and empirical risk reduction is inferred from uncertainty sampling without measuring loss trajectories on queried samples. As a result, the framework functions more as a post-hoc interpretation of known AL behavior rather than a predictive and testable theory. The causal link between the four generalization components and observed performance remains largely speculative.
(W2) Lacks ablation studies; moreover, the baseline methods are not directly tied to the core arguments of the paper. The baseline results cannot directly demonstrate the performance or the roles of the four key arguments proposed in the paper.
(W3) Experiments rely on outdated datasets (CIFAR, small ImageNet subsets).
(W4) Baseline methods are severely outdated and there’s limited discussion of modern large-scale or multimodal AL scenarios. Together with the reliance on outdated datasets, it limits the applicability of the findings to the latest image classification tasks and recent large models. The scalability of the proposed framework is highly questionable.
(W5) The qualitative visualizations in Figure 1 only indicate where points are selected in the embedding space, which does not itself validate whether such selections are actually effective for improving performance. A behavior may look “reasonable” yet still lead to suboptimal learning outcomes. Robust conclusions also require quantitative evidence, not just visual intuition.

**Questions:**

(Q1) How sensitive are your conclusions to model architecture or embedding dimensionality? Specifically, will the proposed framework reach similar findings when extended to large- or multimodal base models (e.g., CLIP, ViT, LLaVA)?
(Q2) What is the real-world meaning of studying AL under the selected baseline models in this work? Specifically, it is true AL helps reduces annotation cost, but the lack of  studying under recent SOTA models severely limits the practical meaning of this work.

Please explain how you will modify the paper specifically. As reviewers, we are not interested in just private education but in how the manuscript will be improved.

---

> ### Author Response · Authors · 2025-12-01
>
> We sincerely thank the reviewer for the constructive and detailed evaluation. We appreciate the recognition that the paper provides a useful theoretical lens and helps explain previously observed AL phenomena. Below, we address each concern and clarify the intended scope of our contribution.
>
> (Q1) The decomposition is defined at the level of distributions and hypothesis classes, and hence, the four terms (empirical risk, discrepancy, complexity, and confidence) exist regardless of backbone architecture choice (CNN, ViT, CLIP, or LLaVA).  The only changes would be related to the magnitude of the complexity term, the shape of the feature space for AL methods (e.g., TypiClust for better separation of classes with the most representative samples), and the relative dominance of representation vs. coverage. In this study, our focus is on interpreting the generalization terms’ behaviour for different AL strategies under different budgets, datasets, and SSL scenarios using the same backbone architectures for a fair comparison. However, we agree that investigating the architectural differences could be an interesting direction.
>
> (Q2) In many real-world annotation pipelines, such as medical imaging, robotics, and autonomous driving, the number of available unlabeled samples ranges from a few to a few million. However, the annotation budget is typically limited due to expert cost, time constraints, or operational requirements. In these settings, practitioners need to make principled decisions about which AL strategy to deploy under specific budget constraints. Without understanding why certain AL methods succeed or fail under different regimes, we are forced to rely on expensive trial-and-error: running multiple AL strategies, each requiring repeated training cycles, to determine which performs best for their scenario.
> This is often infeasible in high-stakes or time-critical domains.
> By linking AL strategies to the four components of the generalization decomposition, the paper provides:
> (a) A principled explanation of why certain strategies outperform others in different budget regimes.
> (b) Guidance on which AL family to prefer, uncertainty, representativeness, or coverage, given the scale of the unlabeled pool, the expected budget, and the characteristics of the embedding space.
> (c) A reduction in the need for costly empirical sweeps, since we can select a promising AL strategy based on theoretical alignment rather than full experimentation.
> Thus, the proposed decomposition link increases the practical usefulness of AL by helping to make informed decisions under real-world constraints where budgets are small and iterative experimentation is prohibitively expensive.
>
> (W1) We agree that isolating each of the four components, including quantitative trends for all components, would strengthen the paper. We intended to introduce the decomposition as a conceptual framework connecting AL principles to generalization theory. The empirical section, therefore, focused on demonstrating qualitative alignment between the decomposition and the observed behavior of AL methods across budget regimes.
>
> (W2-5) We appreciate the reviewers’ point and agree that the paper does not cover all existing AL baselines; however, the purpose of the paper is not to benchmark SOTA methods but to explain why they behave differently, and the decomposition is fully compatible with both classical and modern AL algorithms. Our choice for baselines and dataset was mainly to show a fair comparison across most known AL methods from different categories and to understand the underlying patterns regarding different generalization components. We agree with the reviewer that the additional ablation studies and quantitative measures of the four generalization components are important, and including new datasets and SOTA AL methods will strengthen the paper and the contribution clearly.

---

### Official Review · Reviewer_DEhC · 2025-10-31

**Soundness:** 2
**Presentation:** 2
**Contribution:** 3
**Rating:** 2
**Confidence:** 4

**Summary:**

This work proposes an analysis framework that connects the traditional probably approximately correct (PAC) learning approach with the active learning (AL) process. Specifically, the authors decompose the generalization error of passive machine learning into empirical risk, model complexity, distributional discrepancy, and confidence terms. Based on this framework, they revisit the common design choice of active learning methods categorized into uncertainty, coverage, and representation.

**Strengths:**

1. They develop the analysis framework from the PAC learning and extend it to the AL framework, which reveals the differences, such as non-i.i.d. sampling and the impact of budget size.
2. They give the empirical results to show that the uncertainty, coverage, and representation play different roles under various budget regimes, which might be useful to review the previous AL algorithms and for future design.

**Weaknesses:**

1. **Need to discuss more related works.** About fitting the generalization error to the AL process, for example, (1) also proposed the unified framework of decomposing the generalization error (expected risk) into the empirical risk and the distribution discrepancy, and further proposed the practical solution for the deep active learning paradigm. Although this work focuses on the analysis framework without proposing a new method, it should also cover more related works to recall previous efforts in this area, such as (2).
2. **Clarify the differences between Representation and Coverage.** While decoupling the terms of diversity-based methods (3) from the representation and coverage is novel, it should differentiate more carefully in Section 2. For example, *Coverage-based methods promote broad exploration of the input space using geometric or probabilistic criteria.* Even if we use probabilistic criteria, why can not we interpret it as a kind of representation-based method? Or, given the coverage aspect, what insights could we gain that are different from the representation-based method?
3. Following 2., *TypiClust as representation-based clustering that selects typical (high-density) and diverse samples; ...; ProbCover that maximizes probabilistic coverage of the feature space*. What is the key difference between TypiClust and ProbCover beyond the different approaches to achieve more coverage of the feature spaces?
4. **Reveal more parts of the experimental settings.** In your experiments, do you retrain the whole ResNet-X or only fine-tune the last or the penultimate layer?
5. Following 4., Figure 1 seems to indicate that you keep the feature extraction layers, i.e., the layers before classifying. It seems obvious that the representation methods explore more in the early stages, given this setting. However, what if we retrain whole networks during the AL process? Does the observation still hold for the representation methods?
6. **Provide more derivation of Eq. (11).** It is unclear to me how to obtain the *Model Complexity* and *Confidence term* given the previous Sections. Could you provide more explanations? For example, the probabilistic sample complexity tells us that $m = \mathcal{O}(d, \delta, \epsilon)$. Why does the dimension $d$ disappear in Eq. (11) of the Confidence term?

- (1) Shui, C., Zhou, F., Gagné, C., & Wang, B. (2020, June). Deep active learning: Unified and principled method for query and training. In International conference on artificial intelligence and statistics (pp. 1308-1318). PMLR.
- (2) Wang, Z., & Ye, J. (2015). Querying discriminative and representative samples for batch mode active learning. ACM Transactions on Knowledge Discovery from Data (TKDD), 9(3), 1-23.
- (3) Ren, P., Xiao, Y., Chang, X., Huang, P. Y., Li, Z., Gupta, B. B., ... & Wang, X. (2021). A survey of deep active learning. ACM computing surveys (CSUR), 54(9), 1-40.

**Questions:**

1. For the Model Complexity, you refer to (4) (I guess that you refer to Sec 4.2?) to measure the empirical Rademacher complexity as the Eq. (14). However, their design for the neural networks is only explained for the two-layer neural networks. How could we ensure that the measurement still holds for the ResNet or other types of convolutional neural networks (CNNs)?

- (4) Bartlett, P. L., & Mendelson, S. (2002). Rademacher and gaussian complexities: Risk bounds and structural results. Journal of machine learning research, 3(Nov), 463-482.

---

> ### Author Response · Authors · 2025-12-01
>
> We sincerely thank the reviewer for the constructive and detailed feedback. We are pleased that the reviewer found the framework and empirical insights useful. We appreciate the reviewer’s suggestion and address each point below.
>
> (Q1) Our model-complexity term in Eq. (11) is the empirical Rademacher complexity Rad_S(l o H), defined in Eq. (14). The reference to Bartlett and Mendelson (2002) is used for the general Rademacher-based generalization bound, not for their specific construction for two-layer networks. The definition of Rad_S(l o H) applies to any hypothesis class H, including deep CNNs such as ResNet, and our empirical estimator is computed directly from the trained model’s predictions on the labeled sample. We therefore do not assume the specific architectural setup of their two-layer example. More refined upper bounds on Rad_S(l o H) for deep networks (e.g., via spectral norms and layerwise products) are indeed available in later work, but they would only change the numerical form of the complexity term and are orthogonal to our main goal, which is to isolate and compare the contributions of empirical risk, distribution shift, model complexity, and confidence across AL strategies.
>
> (1) We agree that prior work on decomposing generalization error, particularly the decomposition into empirical risk and distributional discrepancy, is relevant to our goal. However, our intention in the submission was to focus on adapting a PAC-style decomposition specifically to the active learning setting, where non-i.i.d. selection induces shifts in the unlabeled pool, the labeled pool, and the induced hypothesis trajectory. This setting differs from prior decompositions in two important ways:
> (a) Selection-induced support shift: unlike passive learning, AL actively changes the support of the empirical distribution across rounds.
> (b) Budget/regime-dependent behavior: empirical risk, coverage, and representation terms evolve differently depending on the labeling budget, which is central to our analysis.
>
> (2) We agree that this distinction should be sharpened. The term coverage is indeed overloaded in the literature, as it may refer to:
> (a) representation coverage: density or typicality in feature space
> (b) geometric coverage: spatial dispersion of selected points
> (c) probabilistic coverage: mass coverage under the data distribution
> In our framework, Representation corresponds to density-based selection, choosing points that lie in typical or high-density regions of the feature space.
> Coverage, in contrast, refers to geometric expansion of the labeled set, selecting samples that increase the spatial support of labeled points and thereby reduce the discrepancy between the labeled-set distribution and the underlying data distribution.
> This distinction allows us to separate “sampling near the center of mass” (representation) from “sampling to expand support” (coverage), which aligns with how these principles manifest in AL practice.
>
> (3) Thank you for raising this point. We agree that the conceptual differences can be made clearer.
> (a) TypiClust selects cluster-typical samples (high-density points after clustering). These are representations of dominant modes in the feature space.
> (b) ProbCover does not rely on clustering. Instead, it computes the local density in a continuous neighborhood and selects points in regions whose mass contributes most to global distribution coverage. Unlike TypiClust, ProbCover’s objective is explicitly tied to probabilistic coverage rather than to typicality within a cluster.
> Thus, while both methods relate to representation in a broad sense, their operational principles differ: TypiClust emphasizes within-cluster typicality, while ProbCover emphasizes distributional mass coverage without assuming cluster structure.
>
> (4) In all experiments, we retrain the full ResNet-18/50 from scratch at every AL round, rather than fine-tuning only the last or penultimate layer. This choice follows the protocol of Hacohen et al. (2022) and isolates the effect of the queried set rather than accumulating optimization artefacts across rounds.
>
> (5) For t-SNE visualizations, we extract features from the penultimate layer of a network trained only on the currently labeled set.
> We use these features solely for illustrating where different AL methods sample, visualizing cluster structure, and explaining qualitative differences between selection behaviors.
> For representation-based methods such as TypiClust, this layer is also where clustering occurs.
> For uncertainty-based or coverage-based methods, the t-SNE plot is purely illustrative and does not drive training.
> Importantly, the selection step is independent of the visualization, and once the samples are selected, the model is retrained from scratch using supervised learning on the updated labeled set. Thus, the observations about representation vs uncertainty vs coverage remain valid.

---

> > ### Author Response · Authors · 2025-12-01
> >
> > (6) Thank you for pointing out that the derivation of Eq. (11) was too compressed. The equation is obtained by combining a standard Rademacher-complexity generalization bound with a domain-adaptation/IPM decomposition. For a bounded alpha-Lipschitz loss l, Bartlett and Mendelson (2002, Theorem 7) show that, for any distribution P and any sample S of size m drawn i.i.d. from P, with probability at least 1 - delta we have for all h in H:
> > R_P(h) <= R_S(h) + 2 * Rad_S(l o H) + alpha * sqrt( 2 * log(4/delta) / m )
> > We then follow Menden et al. (2025) in decomposing the AL risk under the sampling distribution Q into the risk under the test distribution P_X plus an IPM term d_F(P_X, P_Q), which yields Eq. (11).
> > Regarding your comment that “probabilistic sample complexity tells us m = O(d, delta, epsilon),” the dependence on the dimension (or VC-dimension) d is captured by the model complexity term Rad_S(l o H). For many hypothesis classes, one has Rad_S(l o H) = O( sqrt(d/m) ), and substituting this into Eq. (11) recovers the usual sample-complexity statement. In our decomposition, we intentionally keep Rad_S(l o H) explicit and treat the last term alpha * sqrt( 2 * log(4/delta) / m ) as a dimension-free “confidence” term that depends only on the failure probability delta and the sample size m.

---

### Meta-Review · Area_Chair_RxUF · 2025-12-18

**Summary:**

This paper studies several aspects of active learning. While the problem is important, reviewers found that the technical contribution is limited and there is disconnection between theory and experiments.

**Reviewer Concerns:**

I do not believe that through rebuttal, technical weakness can be addressed.

**Reviewer Scores:**

Reviewers are unlikely to change scores.

---

### Decision · Program_Chairs · 2026-01-26

Reject